# An Improved Gas Leakage Model and Research on the Leakage Field Strength Characteristics of R290 in Limited Space

**Yalun Li, Peixu Zhou, Yuan Zhuang, Xilei Wu, Ying Liu, Xiaohong Han * and Guangming Chen**

Key Laboratory of Refrigeration and Cryogenic Technology of Zhejiang Province, Institute of Refrigeration and Cryogenics, Zhejiang University, Hangzhou 310027, China; 12027105@zju.edu.cn (Y.L.); 22027101@zju.edu.cn (P.Z.); zhuang_yuan@zju.edu.cn (Y.Z.); lu_wxl@zju.edu.cn (X.W.); 12127080@zju.edu.cn (Y.L.); gmchen@zju.edu.cn (G.C.)

\* Correspondence: hanxh66@zju.edu.cn

**Abstract:** Some alternative refrigerants with excellent environmental performance often have different flammable limits. When refrigerant leaks, the external space may have a certain explosion risk if the refrigerant is not diffused timely. To understand the leakage and diffusion characteristics of the refrigerant, an improved gas leakage model was proposed in this paper, and the accuracy verification of the improved model was developed. Based on the above works, taking R290 as the research object, the variation law of the field strength between the leaked gas and external space and the influence of different initial leakage pressures on the field strength characteristics were analyzed. The simulation results showed that when the initial leakage pressure was 2 MPa, the R290 gas entered the external space as a supersonic jet, the gas underwent continuous expansion and compression processes in the near-field area and a Mach disk was formed within the flow area. During this process, parameters, such as the temperature, pressure, velocity and density of the leaked R290 gas, changed dramatically, and then the gas gradually returned to room temperature and normal pressure through interaction with the external space. The flammable area formed by the leaked R290 was mainly concentrated in the local flow area below the leak hole, and the existence of the Mach disk caused the R290 high concentration area to increase. With the increase in the initial leakage pressure, the distance from the Mach disk to the leak hole and the circumferential diameter of the Mach disk increased, and the flammable area increased slightly in the horizontal direction, whereas the flammable area increased significantly in the vertical direction.

**Keywords:** improved gas leakage model; R290; field strength characteristics; Mach disk; flammable area

## 1. Introduction

Massive commercial refrigeration systems are highly susceptible to having loose components and refrigerant leakages. These systems have many complex pipes. Any improper installation, maintenance or tightening can result in loose or broken fittings, which weaken connection points and allow for leaks [1]. The mechanics of a refrigerator, which operates by expanding and contracting refrigerant under pressure, adversely affects the integrity of seals. In fact, vibration and regular wear and tear, itself, can exacerbate leakages. For these reasons, it is nearly impossible to have a system that will never leak in its lifetime. Refrigeration and air conditioners have a very wide range of applications around the world. They usually contain mechanical moving parts, long pipelines, small parts, etc. During long-term operation, due to the continuous vibration of pipelines, friction and corrosion of working fluids, weathering of external space, or external forces, etc., cracks, holes or gaps often appear in the pipeline, which in turn lead to the occurrence of refrigerant leakage events [2,3]. Beshr et al. [4] found that the annual leakage rate of the commercial refrigeration system of a supermarket chain in the United States was close to 30%. Kim et al. [5] mentioned that among the more than 4000 residential air conditioners surveyed in California, the refrigerants leaked from approximately 62% of the

systems. When these high GWP (global warming potential) gases and, in some instances, still ODS (ozone depleting substance), inevitably leak, they pose a direct threat to our climate and environment. Moreover, with the surge in the application of refrigeration and air-conditioning equipment year by year, the impact of leaked refrigerant on the climate increases significantly [6]. Therefore, the replacement of refrigerants should be further implemented to reduce the application of refrigerants with high GWP. Refrigerants with potential to replace high GWP hydrofluorocarbons (HFCs) mainly include natural working fluids, hydrofluoroolefins (HFOs), ethers (HFEs), low GWP HFCs and mixed refrigerants, and some alternative refrigerants have different flammable limits [7]. When the flammable refrigerant leaks from the refrigeration system to the external space, if it is not diffused in time, the concentration of the leaked refrigerant in the local area exceeds its lower flammable limit (LFL) and a flammable area is formed, which threatens the safety of the external space [8]. Therefore, in order to promote the application of alternative refrigerants with superior environmental performance, a large number of theories on the leakage characteristics, the diffusion process and concentration distribution characteristics of flammable alternative refrigerants in local areas have been carried out.

The hole model and pipeline leakage model, established by Levenspiel and Crowl et al. [9,10] to describe the gas leakage in pipelines, appeared earlier, which could be used to calculate the leakage and diffusion characteristics for a very small hole area and a completely broken pipeline, respectively. Later, Montiel et al. [11] established a more general medium/low pressure pipeline gas leakage model, also known as the big-hole model. On the basis of the three models, some revised models were proposed to further improve their calculation accuracy for gas leakage and diffusion process. For example, Yang et al. [12] analyzed the heat loss in the gas flow process in the pipeline, and a steady-state leakage model under non-isothermal conditions was established, and the gas temperature at the center point within the pipeline above the leak hole was closer to the actual temperature. Jo et al. [13] introduced a correction factor for pressure drop due to the frictional losses along the pipeline and proposed a simplified model for estimating the leakage rate of high-pressure gas pipelines, which slightly overestimated the leakage rate. Woodward et al. [14] established an unsteady leakage model by adding a time term to the hole model, and the research found that most of the gas in the pressure vessel leaked in the critical flow state, and the average leakage rate of the gas in the critical flow process could be used to approximately describe the entire leakage process. Singh et al. [15] proposed to apply the Adomian decom-position method to solve the one-dimensional unsteady adiabatic gas dynamics equations established in the unsteady leakage model, and the approximate velocity, density and pressure distributions could be obtained. Olorunmaiye et al. [16] established an unsteady isothermal pipeline leakage model to calculate the gas leakage process in the pipeline, which was completely cut off, and the research showed the leakage flow rate obtained by the improved model was 18% lower than that of the hole model. Moloudi et al. [17] proposed a dimensionless leakage correlation by analyzing the effects of the initial leakage pressure, pipe wall friction coefficient and leak hole diameter on the leakage rate, and the accuracy of the correlation was further verified under different leakage sizes and boundary conditions.

Related theoretical research on the diffusion process of refrigerants leaking from different types of systems into the external space has also been developed. For example, Colbourne et al. [18] studied the concentration distribution of CHs refrigerants (i.e., R290, R600 and R600a) leaked from the indoor unit shells of different types of household split air conditioners into the external space. Colbourne et al. [19,20] established a mathematical model to obtain the minimum airflow velocity when the leaked refrigerant was fully diluted into the external space, and the diffusion behaviors of R32, R290 and R1234yf at different airflow rates in an indoor unit of a wall-mounted air conditioner were discussed. Cai et al. [21] studied the diffusion process and concentration distribution of R290 after it leaked from a wall-mounted air conditioner into three rooms with different sizes; the results found that the larger space was more conducive to the dilution and uniform distribution of

the leaked refrigerant. Prakash et al. [22] analyzed the effects of leakage rate, indoor unit installation height, whether an indoor fan was on or off, and the type of furniture placed in the external space on the concentration distribution of leaked R290 in a wall-mounted air conditioner. Hu et al. [23] studied the influence of the airflow rate and angle of the indoor unit on the diffusion process of R290 in a wall-mounted air conditioner, and the results showed that the airflow rate could dominate the diffusion of the leaked refrigerant, and the airflow angle only affected the dispersion when the velocity was small, and upward blowing was more conducive to uniform distribution of the refrigerant. He et al. [24] simulated the influence of obstacle types and placements in the external space on the distribution of refrigerant concentration, and the results showed that if the furniture was placed under the air conditioner, the leaked refrigerant accumulated in the semi-enclosed area enclosed by the furniture and nearby walls, resulting in the formation of a flammable area, and furniture in other locations had less influence on the refrigerant diffusion process. Li et al. [25] studied the diffusion process of R32 after leaking from wall-mounted and cabinet air conditioners, and the simulation results found that the refrigerant leaked from the cabinet air conditioner was not easily diffused to higher places, and it sank to the ground quickly, and the refrigerant had an obvious sedimentation effect when it leaked from the wall-mounted air conditioner. Elatar et al. [26] simulated the change of concentration distribution and flammable volume of the leaked R32 in a rooftop air conditioner under different charges, and the results showed that the concentration of R32 in most areas of the air supply duct was in the flammable area, and the leaked refrigerant in the room mainly appeared under the air supply port closest to the leak hole, and the duration of the flammable area in the room was prolonged with the increase in the refrigerant charge. Liu et al. [27–29] studied the influence of leakage rate, room size, ventilation intensity, and the locations of air inlet and outlet on the concentration distribution of leaked R290 in the multiline air conditioner, and the results found that the volume of the flammable area in the room increased with the increase in the leakage rate, and the mechanical ventilation well reduced the concentration of leaked gas in most areas of the room to below its LFL, and the lower the position of the exhaust port, the faster the refrigerant concentration in the room decreased. Colbourne et al. [30] simulated the influence of refrigerant charge, leakage rate, leakage direction, leakage height, indoor unit shell, space airflow and obstacles on the concentration distribution of leaked R290 in a commercial refrigeration cabinet. Oreilly et al. [31,32] evaluated the risk of refrigerant leakage in the refrigerated transport containers and conducted leakage experiments in different scenarios. Zhen et al. [33] studied the leakage characteristics of a portable refrigerated display cabinet, and the results found that the leaked refrigerant was mainly concentrated in the area near the ground below the leak hole, and the cabinet placed in the corners caused the flammable zone to last longer.

It can be seen from the above literature review that the refrigerant leakage based on the refrigeration system was approximately simplified to the leakage of refrigerant gas in the pipeline. The hole model was generally used to calculate the leakage process. In the hole model, the pressure and temperature of the gas at the junction of the leakage hole and the outer wall of the pipe were directly assumed, which were equal to the ambient temperature and standard atmospheric pressure in the subsonic flow state; in addition, the influence of the change in the refrigerant thermo-physical parameters on the diffusion process of refrigerant gas leaked from the pipeline to the external space was ignored. Actually, this assumption and ignorance had a certain effect on the concentration field of the leaked gas obtained in the external space, especially for flammable gases. This effect may lead to the inability to accurately determine the scope of the flammable area, which brings potential safety hazards. Based on the current research status, the main goal of this paper was that an improved gas leakage model was proposed and its reliability was discussed; taking R290 as the object, the variation law of field strengths (i.e., temperature field, pressure field, velocity field and concentration field) of the leaked R290 gas was analyzed, and the mutual

coupling characteristics between multiple field strengths and the influence of different initial leakage pressures on the leakage gas field strength were further discussed.

## 2. The Improved Gas Leakage Model of Pressure Pipeline

### 2.1. The Hole Models for Gas Leakage

The refrigerant leakage characteristics in refrigeration systems are usually analyzed by means of the hole model in many studies. In the hole model, the leakage process of gas in the pipeline is shown in Figure 1. The process zero to one is the gas transportation process from the steady state point to the leak hole, and the process one to two is the gas leakage process from the inside of the pipeline to the external space through the leak hole. The flow state of the gas at the leak hole is considered, which is divided into sonic flow and subsonic flow based on the critical pressure ratio (CPR) [9]. For the different flow states of the gas in the leakage section, the corresponding solution methods are given in the hole model, and their expression are as follows:

$$
\begin{cases}
CPR = \dfrac{p_a}{p_{cr}} = \left(\dfrac{2}{k+1}\right)^{\frac{k}{k-1}} \\[2mm]
Q = \xi A_{or} p_1 \sqrt{\dfrac{kM}{RT_1}\left(\dfrac{2}{k+1}\right)^{\frac{k+1}{k-1}}} \qquad p_1 \geq p_{cr} \text{ The gas is in sonic flow state} \\[2mm]
Q = \xi A_{or} p_1 \sqrt{\dfrac{2k}{k-1}\dfrac{M}{RT_1}\left[\left(\dfrac{p_a}{p_1}\right)^{\frac{2}{k}} - \left(\dfrac{p_a}{p_1}\right)^{\frac{k+1}{k}}\right]} \qquad p_1 < p_{cr} \text{ The gas is in subsonic flow state}
\end{cases}
$$

(1)

where $p_a$ is the atmospheric pressure of the surrounding environment, MPa; $p_{cr}$ is the critical pressure, that is, the corresponding pressure at the point 1 when the gas in the leakage section changes from subsonic flow to sonic flow, MPa; $p_1$ is the pressure inside the pipeline, MPa; $Q$ is the mass flow rate of the leaking gas, kg/s; $A_{or}$ is the leak hole area, m$^2$; $\xi$ is the orifice loss coefficient; $k$ is the isentropic index; $M$ is the kilomolar mass of the gas, kg/kmol; $R$ is the gas constant, kJ/(kmol·K).

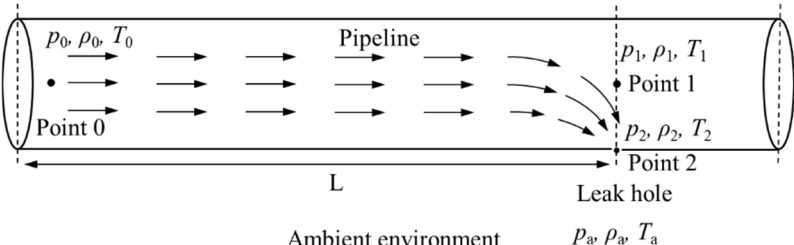

**Figure 1.** Schematic diagram of the gas leakage process in the hole model.

The hole model reveals the leakage mechanism of the gas in the pipeline to a certain extent, but there are still limitations: (1) the change of the gas state parameters in the interval from the leakage port in the pipeline to the outer wall of the pipeline during the gas leakage process is ignored in the modeling; (2) the gas flow state in the leakage section is directly determined according to the given critical pressure ratio formula, and the pressure of the gas at the outlet of the leakage section is assumed. For example, when $p_1 < p_{cr}$, the pressure of the gas at the leak hole is equal to the atmospheric pressure, $p_2 = p_a$; when $p_1 \geq p_{cr}$, the pressure of the gas at the leak hole is equal to $p_1 \cdot CPR$, $p_2 = p_1 \cdot CPR$. The above limitations will lead to certain deviations in the calculated leakage mass flow rate and state parameters of the leaked gas.

### 2.2. The Improved Gas Leakage Model and Its Corresponding Derivation

The improved gas leakage model was proposed based on the above hole model, and its detailed derivation was as follows. The leakage of gas in the pipeline was considered in three stages. The schematic diagram is shown in Figure 2. Zero to one is the gas transportation process in the pipeline. The flow direction of the gas was considered to

be parallel to the axis of the pipeline, and the heat transfer between the leakage gas and the external space was ignored, but the effect of the frictional resistance between the gas and the pipeline wall cannot be ignored. This process was treated as an adiabatic process. One to two is the movement process of the gas inside the pipeline to the leakage hole. The gas streamline changes due to the pressure differences inside and outside the leakage section. At the same time, the change in the flow area makes the temperature and pressure of the leaked gas decrease and the velocity of the leaked gas increase, which can be called a throttling process. In this process, the viscous resistance, orifice loss and heat transfer between the gas and the pipeline wall can be approximately ignored. Thus, it was solved as an isentropic process. Two to three is the secondary expansion process of the refrigerant gas in the leakage section. Because the initial leakage pressure of the refrigerant in the refrigeration system is high, the pressure of the gas in the leakage section is still higher than the atmospheric pressure after the throttling process, and the refrigerant gas is in an under-expanded state (The gas with a certain initial pressure leaks or sprays from the pipeline or container, and the pressure of the gas at the nozzle outlet or the outlet of the leakage section is still higher than the ambient pressure, which is called insufficient expansion, also known as the under-expansion state). When the under-expanded gas is close to the outlet of the leakage section, it will expand further. In the improved model, the gas transportation, throttling and expansion process were regarded as a one-dimensional motion along the gas flow direction, the leakage gas was regarded as the compressible fluid and the ideal gas state equation was adopted. The compression factor Z was introduced to reduce the deviation caused by the calculation of the actual gas leakage.

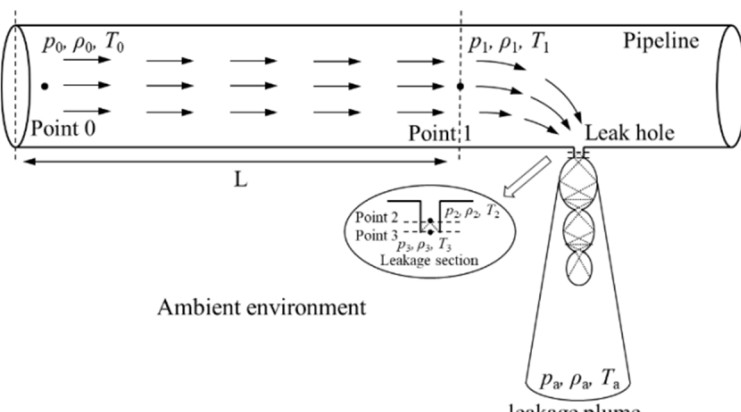

**Figure 2.** Schematic diagram of the gas leakage process in the improved model.

The gas transport process zero to one can be described by the energy and momentum equations, respectively, and the gas motion equation can be obtained by integrating the established equations:

$$\frac{k+1}{k} \ln\left(\frac{p_0 T_1}{p_1 T_0}\right) + \frac{M}{RG^2}\left(\frac{p_1^2}{T_1} - \frac{p_0^2}{T_0}\right) + \frac{4fL_e}{D} = 0 \qquad (2)$$

where $G$ is the mass flow rate per unit cross-sectional area of the pipe, kg/(m²·s); $f$ is the friction coefficient of the pipe wall; $L_e$ is the effective length of the pipe, m; $D$ is the pipe diameter, m; $k$ is the isentropic index of the leakage gas; $M$ is the kilomolar mass of the leakage gas, kg/kmol; $R$ is the gas constant, kJ/(kmol·K).

Introducing the sound velocity $c = \sqrt{kRT}$ and Mach number $Ma = \frac{v}{c}$ into the energy conservation equation, assuming that $f$ is a constant value during the flow of the gas in the

pipeline, the relationship between the gas state parameters at points zero and one can be obtained by solving the equations.

$$
\begin{cases}
\frac{T_1}{T_0} = \frac{Y_0}{Y_1} \\
\frac{\rho_1}{\rho_0} = \frac{Ma_0}{Ma_1} \sqrt{\frac{Y_1}{Y_0}} \\
\frac{p_1}{p_0} = \frac{Ma_0}{Ma_1} \sqrt{\frac{Y_0}{Y_1}}
\end{cases}
\tag{3}
$$

where $Y_i = 1 + \frac{k-1}{2}Ma_i^2$.

Bringing the state parameters of the gas at points zero and one into the gas motion Equation (2), the relationship between the Mach number and frictional resistance during adiabatic gas flow can be obtained:

$$
\frac{k+1}{2k} \ln\left(\frac{Ma_1^2 Y_0}{Ma_0^2 Y_1}\right) - k\left(\frac{1}{Ma_0^2} - \frac{1}{Ma_1^2}\right) + \frac{4fL_e}{D} = 0
\tag{4}
$$

Combined with the parameters of the gas at steady state point 0 and Equations (3) and (4), the state parameters of the gas at point 1 can be obtained.

For the throttling process one to two of the gas in the pipeline, it is described by the continuity equation, the energy and momentum differential equation, the ideal gas state equation and the isentropic process equation. Differentiating the continuity equation, the gas state equation and the logarithm of the isentropic process equation, respectively, the relationship between the gas parameters and the channel cross-section can be obtained by combining with the momentum differential equation:

$$
\begin{cases}
\frac{dA}{A} = \left(Ma^2 - 1\right)\frac{dv}{v} \\
\frac{dp}{p} = \frac{kMa^2}{1-Ma^2}\frac{dA}{A} \\
\frac{d\rho}{\rho} = \frac{Ma^2}{1-Ma^2}\frac{dA}{A} \\
\frac{dT}{T} = \frac{Ma^2(k-1)}{1-Ma^2}\frac{dA}{A}
\end{cases}
\tag{5}
$$

The energy equation is integrated and brought into the state parameters of the gas at points 1 and 2, combined with the ideal gas state equation and the continuity equation, the calculation formula of the leakage gas mass flow rate can be obtained:

$$
Q = \sqrt{\frac{M}{ZR}\frac{2k}{k-1}\frac{T_1 - T_2}{\left(\frac{T_2}{p_2 A_2}\right)^2 - \left(\frac{T_1}{p_1 A_1}\right)^2}}
\tag{6}
$$

For the secondary expansion process two to three, ignoring the effect of the viscous resistance and heat transfer between the gas and the external space, it is regarded as an adiabatic process and described by the continuity equation and the momentum equation, and the relationship between the gas state parameters can be obtained:

$$
\begin{cases}
\frac{v_3}{v_2} = 1 - \frac{1}{kMa_2^2}\left(\frac{p_3}{p_2} - 1\right) \\
\frac{v_3}{v_2} = 1 + \frac{1}{1-kMa_2^2}\left(\frac{T_3}{T_2} - 1\right)
\end{cases}
\tag{7}
$$

If the transportation distance, L, of the leaked gas in the pipeline is small, the gas transportation process (zero to one) and the leakage process (one to two) can be combined into a process (zero to two), which is considered to be the isentropic expansion process, and the initial velocity of the gas at point zero is assumed to be zero. Based on the above conditions, the relationship between the leakage rate and the gas state parameters and the

Mach number can be obtained. The relationship between the gas state parameters and the Mach number can be expressed by the following equation:

$$
\begin{cases}
\frac{T_2}{T_0} = \frac{1}{Y_2} \\
\frac{p_2}{p_0} = \left(\frac{1}{Y_2}\right)^{\frac{k}{k-1}} \\
\frac{\rho_1}{\rho_0} = \left(\frac{1}{Y_2}\right)^{\frac{1}{k-1}}
\end{cases}
\tag{8}
$$

The relationship between gas leakage mass flow rate and the Mach number can be expressed by the following equation:

$$
Q = A_{or} p_0 Ma_2 \sqrt{\frac{kM}{ZRT_0}\left(\frac{1}{1+\frac{k-1}{2}Ma_2{}^2}\right)^{\frac{k+1}{k-1}}}
\tag{9}
$$

It can be seen from Equations (8) and (9) that as the gas pressure $p_2$ in the leakage section decreases, the Mach number of the gas $Ma_2$ and the leakage mass flow rate increase. Since the cross-sectional area of the leakage section is assumed to be constant, the maximum Mach number of the gas in the leakage section can only reach 1, $Ma_2 = 1$. At this time, the gas in the leakage section is in a sonic flow state, and the leakage mass flow rate also reaches the maximum value. The maximum leakage mass flow rate can be expressed by the following formula:

$$
Q_{\max} = A_{or} p_0 \sqrt{\frac{kM}{ZRT_0}\left(\frac{2}{k+1}\right)^{\frac{k+1}{k-1}}}
\tag{10}
$$

where $Q_{\max}$ is the maximum leakage mass flow rate, kg/s.

When the gas in the leakage section is in a sonic flow state, the corresponding state parameters of the gas are called critical parameters including the critical pressure, $p_{cr}$; the critical temperature, $T_{cr}$; the critical density, $\rho_{cr}$. Bringing $Ma_2 = 1$ into Equation (8), the relationship between the critical flow parameters of the gas in the leakage section and the initial state parameters can be obtained:

$$
\begin{cases}
\frac{T_{cr}}{T_0} = \frac{2}{k+1} \\
\frac{p_{cr}}{p_0} = \left(\frac{2}{k+1}\right)^{\frac{k}{k-1}} \\
\frac{\rho_{cr}}{\rho_0} = \left(\frac{2}{k+1}\right)^{\frac{1}{k-1}}
\end{cases}
\tag{11}
$$

Therefore, it can be seen that the leakage mass flow rate of the gas in the pipeline is related to the initial leakage pressure $p_0$, the $k$ value of the leakage gas, and the Mach number at point 2 $Ma_2$, and $Ma_2$ is determined by the ratio of the gas pressure in the leakage section $p_2$ to the $p_0$. After the initial leakage pressure and the type of leakage gas are determined, there is a critical pressure $p_{cr}$. When $p_2$ is greater than $p_{cr}$, the gas in the leakage section is in a subcritical flow state. When $p_2$ is equal to $p_{cr}$, the gas in the leakage section is in a sonic flow state, and the leakage mass flow rate reaches the maximum value. However, the gas pressure $p_2$ in the leakage section is mainly determined by the geometric parameters of the leakage section.

The comparison of the current hole model and the improved gas leakage model is shown in Table 1, and the advantages of the improved leakage model are summarized as follows: the more accurate that the leakage mass flow rate and leakage gas state parameters can be obtained under all hole leakage conditions, the Mach disk formed by the leaked gas in the local flow area of the external space can be found; the range of the flammable area in the external space can be accurately predicted to a certain extent when the flammable refrigerant leaks.

**Table 1.** Comparison of the current hole model and the improved leakage model.

| Model Leakage Process | | The Current Hole Model | The Improved Gas Leakage Model |
|---|---|---|---|
| | The schematic figure |  |  |
| Similarity | The transport process | The same model and solution method were used. | |
| Difference | The leakage process | The gas pressure at the outlet of the leakage section was directly assumed based on the given critical pressure ratio formula. | The relationship between the gas flow parameters and the flow section was established, and the gas state parameters at point two can be solved. |
| | The expansion process | The change in the gas state parameters in the interval from the leakage port in the pipeline to the outer wall of the pipeline was ignored. | The expansion process of the gas in the leakage section was considered, and the state parameters of the gas at the outlet of the leakage section can be obtained. |

### 2.3. The Numerical Simulation

2.3.1. Governing Equations of the Diffusion Process

The actual diffusion process of the leakage gas in the external space can be considered as a single-phase, turbulent flow and heat transfer process, thus, the mass, momentum and energy conservation equations were adopted. At the same time, the leaked gas was continuously mixed with the air in the external space, and the diffusion flux of the leaked gas was calculated by the component transport equation. The general form of the governing equations is as follows:

Continuity equation:

$$\frac{\partial \rho}{\partial t} + \frac{\partial}{\partial x_j}(\rho u_j) = 0 \tag{12}$$

where $\rho$ is the density of mixed gas composed of leaked gas and air; $x_j$ is the three diffusion directions x, y and z of the leaked gas in external space; $u_j$ is the corresponding velocity component of the three directions.

Momentum equation:

$$\frac{\partial(\rho u_i)}{\partial t} + \frac{\partial}{\partial x_j}(\rho u_i u_j) = -\frac{\partial p}{\partial x_i} + \frac{\partial}{\partial x_j}\left(\mu_t \frac{\partial u_i}{\partial x_j}\right) + \frac{\partial}{\partial x_j}\left(\mu_t \frac{\partial u_j}{\partial x_i}\right) + (\rho - \rho_a)g_i \tag{13}$$

where $\mu_t$ is the turbulent viscosity of the mixed gas; $g$ is the acceleration of gravity; $p$ is the pressure of the mixed gas; $\rho_a$ is the density of the air.

Energy equation:

$$\frac{\partial(\rho T)}{\partial t} + \frac{\partial}{\partial x_j}(\rho u_j T) = -\frac{P}{c_P}\frac{\partial u_j}{\partial x_j} + \frac{1}{c_P}\frac{\partial}{\partial x_j}\left(k_t \frac{\partial T}{\partial x_j}\right) + \frac{c_{P_r} - c_{P_a}}{c_P}\left(\rho D_t \frac{\partial \omega}{\partial x_j}\right)\frac{\partial T}{\partial x_j} + \phi \tag{14}$$

$$\phi = 2\mu_t \left(\frac{\partial u_j}{\partial x_j}\right)^2 + \mu_t \left(\frac{\partial u_i}{\partial x_j} + \frac{\partial u_j}{\partial x_i}\right)^2 \tag{15}$$

where $T$ is the temperature of the mixed gas; $k_t$ is the turbulent thermal conductivity; $c_P$ is the specific heat capacity of the mixed gas; $c_{P_r}$ is the specific heat capacity of the leaked gas; $c_{P_a}$ is the specific heat capacity of the air; $\omega$ is the mass fraction of the leaked gas in the mixed gas.

Component transport equation:

$$\frac{\partial(\rho\omega)}{\partial t} + \frac{\partial}{\partial x_j}(\rho u_j\omega) = \frac{\partial}{\partial x_j}\left(\rho D_t \frac{\partial\omega}{\partial x_j}\right) \tag{16}$$

where $D_t$ is the turbulent mass diffusion coefficient.

### 2.3.2. Geometric Model

The three-dimensional geometric model of the pipeline, the leakage section and the external space was established for the refrigerant leakage and diffusion process in the pressure pipeline as shown in Figure 3. The inner diameter of the pipeline was 6 mm, and the length was 50 mm. The center of the leakage section was 25 mm from the inlet of the pipe. The diameter of the leakage section was 1 mm, the length was 1 mm, and the external space was $2 \times 2 \times 2$ m. In the model, 12 measuring points were set on the XZ plane to monitor the changes in the temperature, pressure and leakage component concentration in the external space during the leakage process, and the coordinates of each point is shown in Table 2.

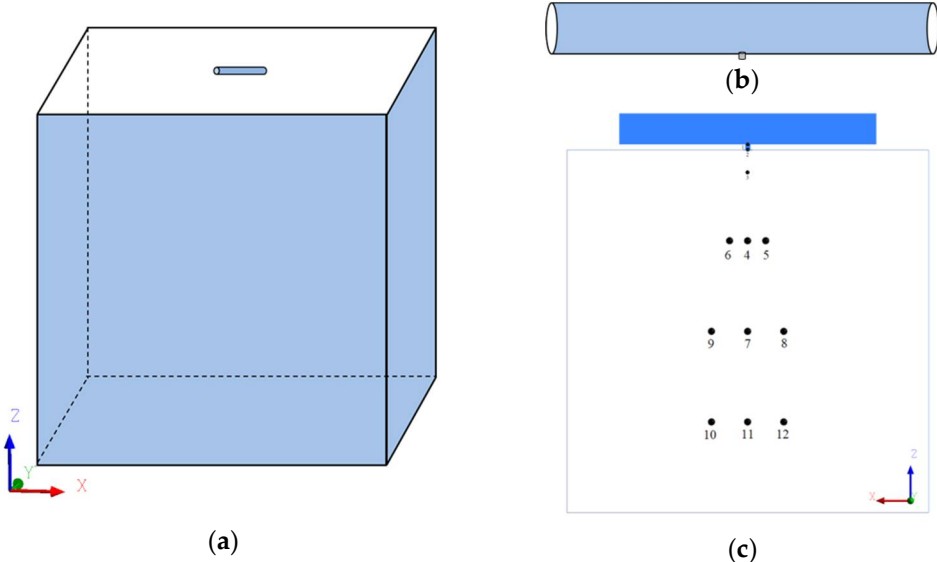

**Figure 3.** Three-dimensional geometric model of gas leakage and diffusion: (**a**) leakage space; (**b**) pipeline and leakage section; (**c**) monitoring surface and distribution of measuring points.

**Table 2.** Coordinates of different measuring point.

| Measuring Point | Coordinates (m) | Measuring Point | Coordinates (m) |
|---|---|---|---|
| point 1 | (0.025, 0, 0) | point 7 | (0.025, 0, −1) |
| point 2 | (0.025, 0, −0.001) | point 8 | (−0.175, 0, −1) |
| point 3 | (0.025, 0, −0.01) | point 9 | (0.225, 0, −1) |
| point 4 | (0.025, 0, −0.5) | point 10 | (0.225, 0, −1.5) |
| point 5 | (−0.075, 0, −0.5) | point 11 | (0.025, 0, −1.5) |
| point 6 | (0.125, 0, −0.5) | point 12 | (−0.175, 0, −1.5) |

### 2.3.3. Numerical Methods and Boundary Conditions

The geometric model was meshed by means of ANSYS software. The whole model adopted the structured hexahedral mesh, and the meshes were locally refined at the inlet of the pipeline, inside the leakage section and at the outlet of the leakage section as shown in Figure 4. The solution method and initial value of several of the parameters in the numerical model are shown in Table 3. For example, the leaked gas had a high velocity inside the leakage section and in the local area of the external space, which may reach

the velocity of sound and had a large Reynolds number, so the SST k-omega turbulence model was adopted. The density term in the model was calculated by the Peng–Robinson equation [34], the viscosity and thermal conductivity were calculated by the mass weighted mixing law model and the mass diffusion coefficient was calculated by the model based on the molecular theory [35]. The pressure inlet was set on the left wall of the pipe, the plane at the bottom of the external space was set as the pressure outlet and the other boundaries were set as walls. A pressure-based transient solver and the coupled algorithm were used to solve the governing equations. The algorithm had great advantages for the free jet of gas in the pipeline and maintained high convergence speed and accuracy.

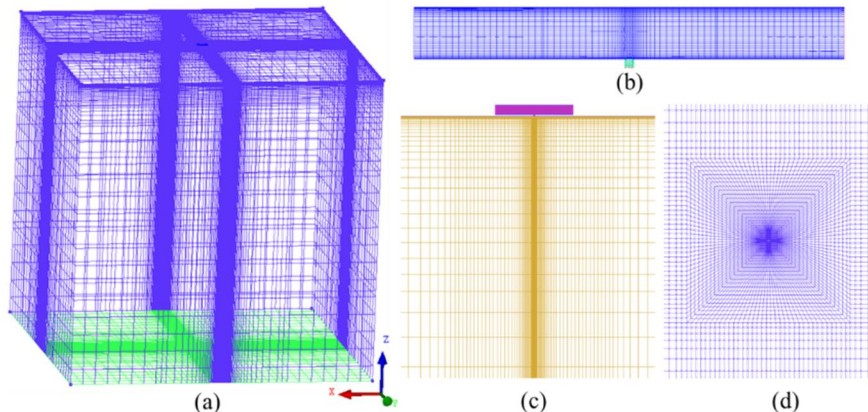

**Figure 4.** Computational meshes of fluid area: (**a**) mesh in the external space; (**b**) mesh in the pipeline and leakage section; (**c**) mesh in the local flow area in space below the leak hole; (**d**) mesh in the XY section.

**Table 3.** The solution method or initial value of the parameters in the numerical model [34,35].

| Numerical Model | Method or Initial Value |
| --- | --- |
| Turbulence model | SST k-omega |
| Leakage gas | R290 |
| Density | $P = \frac{RT}{V-b} - \frac{a}{V^2+2bV-b^2}$ |
| Viscosity | $\mu = \sum_k \omega_k \mu_k$ |
| Thermal conductivity | $k = \sum_{ci} \omega_{ci} k_{ci}$ |
| Diffusion coefficient | $D_{c_1 c_2} = 0.00186 \dfrac{\left[ T^3 \left( \frac{1}{M_{c_1}} + \frac{1}{M_{c_2}} \right) \right]^{1/2}}{P \sigma_{c_1 c_2} \Omega_D}$ |
| Initial leakage pressure $P_0$ (MPa) | 0.5, 1, 1.5 and 2 |
| Initial leakage temperature $T_0$ (K) | 360 |
| Pressure of the external space $P_a$ (kpa) | 101.325 |
| Temperature of the external space $T_a$ (K) | 300 |
| Solver | Pressure-based, transient |
| Solution method | Coupled |

Some assumptions were established for the boundary conditions in the numerical model, as follows:

(1) During the process of refrigerant leakage, compared with the change of the enthalpy value of the leaked gas, the heat transfer between the gas and the walls of the pipe and the leakage section was very small, which was neglected in the numerical model;

(2) When the refrigerant leaked in the refrigeration system, the pressure inside the system changed dynamically with time. To simplify the calculation, the initial leakage pressure was assumed to be a constant value in the model;

(3) In the actual leakage process, the shape of the leakage hole on the surface of the copper tube was very uncontrollable, and the leak holes with different shapes may appear. In the model, a circular hole with a constant cross-sectional area was used as an example to study;

(4) During the actual leakage process, due to the heat transfer between the air and the walls in the external space, the temperature of the space fluctuated, which had a certain impact on the diffusion process of the leaked gas. In the model, the walls were insulated, and the change of external space temperature caused by this part of the heat transfer was ignored.

### 2.4. The Verification of the Improved Gas Leakage Model

When the initial leakage pressures were 1 and 3 Mpa, and the leakage aperture was 1 mm, the experimental research on the hydrogen leakage and hydrogen concentration distribution on the jet axis in the local flow area was conducted by Li et al. [36].

The working conditions of the numerical simulation: the inlet pressures were 1 and 3 Mpa, respectively; the initial temperature of the leaked gas was 300 K; the pressure of the external space was 101.325 kPa, and its temperature was 300 K. The volume fraction of hydrogen inside the pipeline and the leakage section were 1, the external space was air, and the leakage aperture was 1 mm. The comparison between the experimental results from the literature [36] and the numerical results obtained from the hole model and the improved gas leakage model was respectively developed.

#### 2.4.1. Comparison between Numerical and Experimental Results of the Mach Disk Position and Diameter

When the initial leakage pressures were 1 and 3 Mpa, the state parameters of the leaked hydrogen and the leakage mass flow rates calculated by the hole model and the improved leakage model are shown in Table 3. It can be seen from Table 4 and the above description from Section 2.1 that in the hole model, the gas in the leakage section was considered to be in the sonic flow state, and the leakage mass flow rate also reached the maximum value under the above pressure conditions; thus, the Mach disk could not be obtained in the flow area. However, the leakage mass flow rate and the gas pressure at the outlet of the leakage section calculated by the improved leakage model were lower than the calculation results of the hole model, and the leakage velocity at the outlet of the leakage section was higher than the local velocity of sound. The reason for this difference was that when the diameter of the leakage section was 1 mm, the pressure of the gas in the leakage section was higher, and the gas was still in a subsonic flow state, and the leakage mass flow rate had not yet reached the maximum value. When the gas was close to the outlet of the leakage section, the gas underwent a secondary expansion process due to the pressure difference between the inside of the pipeline and the external space, causing its pressure to reduce and velocity to increase. At this time, a choked flow was formed in the leakage section, and the leakage mass flow rate would not increase with the decrease in the gas pressure.

**Table 4.** Leakage gas state parameters obtained from the hole model and the improved leakage model.

| Models | Parameter | Pressure (MPa) | Temperature (K) | Velocity (m/s) |
|---|---|---|---|---|
| | Initial leakage value | 1 | 300 | 0 |
| Hole model | Critical leakage value | 0.53 | 249.32 | 1206.31 |
| | Maximum leakage mass flow (g/s) | | 0.48 | |
| Improved leakage model | Leakage section outlet | 0.40 | 234.32 | 1371.85 |
| | Leakage mass flow (g/s) | | 0.41 | |
| | Initial leakage value | 3 | 300 | 0 |
| Hole model | Critical leakage value | 1.58 | 249.04 | 1213.71 |
| | Maximum leakage mass flow (g/s) | | 1.44 | |
| Improved leakage model | Leakage section outlet | 1.18 | 233.53 | 1381.14 |
| | Leakage mass flow (g/s) | | 1.24 | |

The variation of the Mach number of the leaked hydrogen in the local flow area based on the improved gas leakage model is shown in Figure 5. It can be seen that under the

condition of initial leakage pressures of 1 and 3 Mpa, the Mach numbers of the gas at the outlet of the leakage section were 1.17 and 1.18, respectively. The leaked hydrogen was injected into the external space at supersonic velocity and continued to expand, and a series of expansion waves were formed in the flow area. After the leaked gas through the expansion wave, its temperature and pressure decreased rapidly, the Mach number increased, and the maximum Mach number values reached 3.03 and 4.72 at different pressure conditions, respectively. When the pressure of the leaked gas was lower than the atmospheric pressure, the gas began to be compressed by the surrounding environment and moved toward the direction of the jet axis. At this time, the expansion waves were converted into compression waves, and when countless compression waves met, a normal shock wave was formed in the direction of the vertical gas jet, also known as a Mach disk. After the leaked gas through the Mach disk, the temperature and pressure increased rapidly, the Mach number decreased, and the gas changed from the supersonic flow state to the subsonic flow state. The Mach disk formed by the leaked gas in the flow area was closely related to the gas state parameters at the outlet of the leakage section, so the position and diameter of the Mach disk were selected to verify the accuracy of the gas state parameters calculated by the improved leakage model. Under the condition of 3 Mpa, the distance from the Mach disk to the leak hole obtained by the improved model was approximately 3.4 mm, and the circumferential diameter was approximately 0.7 mm; the two values experimentally obtained by Li et al. in the literature were 3.65 and 0.75 mm, respectively, the deviation between the numerical and experimental values were small. However, based on the calculation results of the hole model and without considering the expansion and compression of the leaked gas, the Mach disk would not be formed in the flow area.

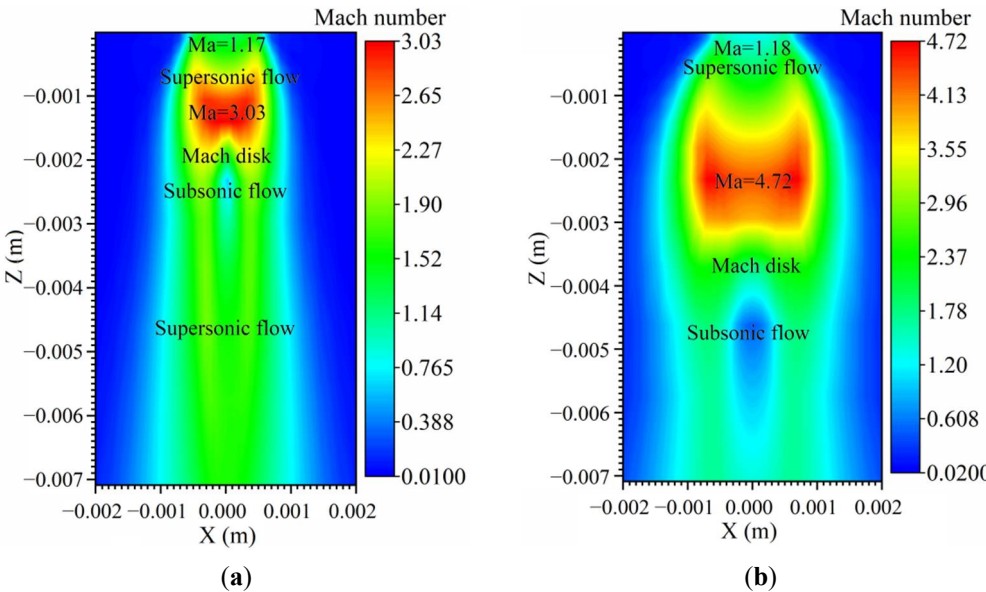

**Figure 5.** Cloud map of the Mach number in the local flow area under different initial leakage pressures: (**a**) initial leakage pressure of 1 MPa; (**b**) initial leakage pressure of 3 Mpa.

2.4.2. Comparison between the Numerical and Experimental Results of the Gas Concentration on the Jet Axis

The comparison between the numerical simulation results obtained by the gas leakage model proposed in this paper and the experimental data of Li et al. [36] is shown in Figure 6. It can be seen from Figure 6 that when the initial leakage pressures were 1 and 3 Mpa, the experimental data and the numerical simulation results were in good agreement, and the concentration deviations of the leaked hydrogen at different positions of the jet axis were within 3 vol%.

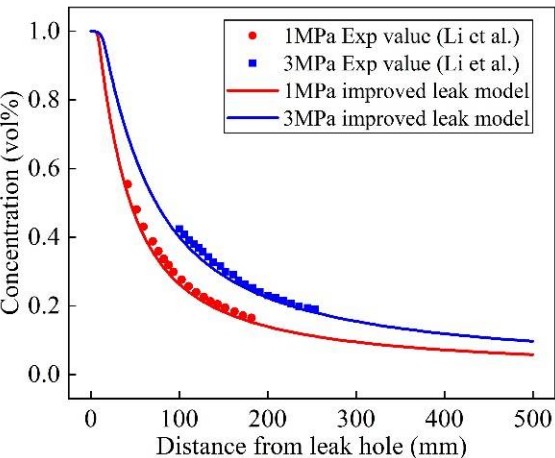

**Figure 6.** Comparison of the experimental and numerical results on the concentration of the leaked hydrogen.

### 3. The Results and Discussion

Propane (R290) is considered to be a potential alternative refrigerant that can be used in small-scale refrigeration systems, but it is extremely flammable (the flammable limit is 2.1–9.5 vol%), which can easily cause safety risks in the event of leakage. The research results on the leakage field strength characteristics of R290 in this paper are as follows.

*3.1. Multi-Field Coupling Characteristics of Leaked R290 in the Near-Field Flow Area*

The initial leakage pressure was 2 Mpa, and the initial temperature was 360 K. The temperature of the external space was 300 K, and the pressure was 101.325 kPa. The temperature, pressure, velocity and concentration distribution curves of the leaked R290 gas on the jet axis in the local area obtained by the improved gas leakage model are shown in Figure 7, and the flow field distribution cloud maps of the leaked R290 in the local area are shown in Figure 8. It can be seen from Figure 7a–c that the temperature and pressure of R290 at the center point of the leakage section outlet were 328.9 K and 0.95 Mpa, respectively. The leaked R290 gas was still in an under-expanded state, and its Mach number was approximately 1.04, and the leakage velocity was greater than the local velocity of sound. After the leaked R290 entered the external space, the temperature and pressure of R290 in the direction of the jet axis decreased rapidly; the minimum values of the temperature and pressure were 199.8 K and 0.013 Mpa, respectively, and the velocity increased rapidly, reaching a maximum of 680.3 m/s. When the temperature and pressure of R290 reached the minimum level, the two parameters rose abruptly, and the gas velocity decreased sharply. After three sharp changes in the state parameters of R290, the pressure of R290 was very close to the atmospheric pressure, the temperature rose to a level slightly higher than the room temperature and the temperature and velocity of the leaked gas gradually decreased along the jet direction. The above phenomena can be explained as follows: the under-expanded R290 gas continued to expand after entering the external space as a supersonic jet. When it expanded below the atmospheric pressure, the leaked gas began to be compressed by the surrounding environment, and a Mach disk was formed inside the flow area. The temperature and pressure of the gas through the Mach disk were compressed to a higher level, and this part of gas then expanded and compressed again. However, with the continuous expansion and compression process, due to the viscous resistance and heat transfer between the leakage gas and external air, the pressure of the leaked gas gradually reduced. Eventually, the expansion and compression processes in the flow area disappeared, and a turbulent cloud with gently changing field strength was formed, and the leaked gas gradually returned to room temperature and atmospheric pressure through heat transfer with the external space.

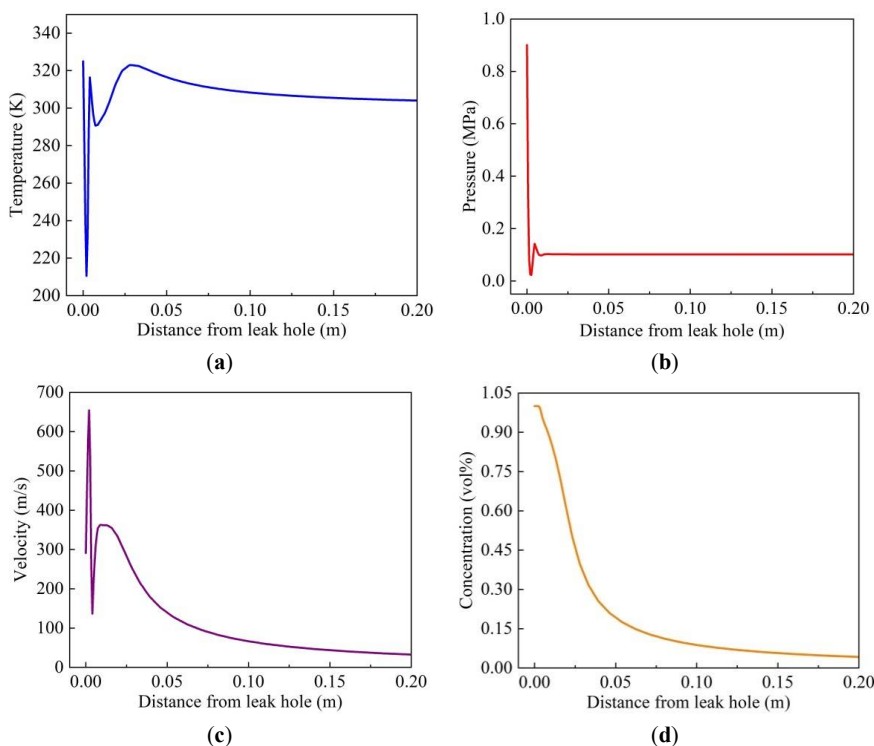

**Figure 7.** Flow field distribution on the jet axis in the local flow area below the leakage hole: (**a**) temperature curve; (**b**) pressure curve; (**c**) velocity curve; (**d**) concentration curve.

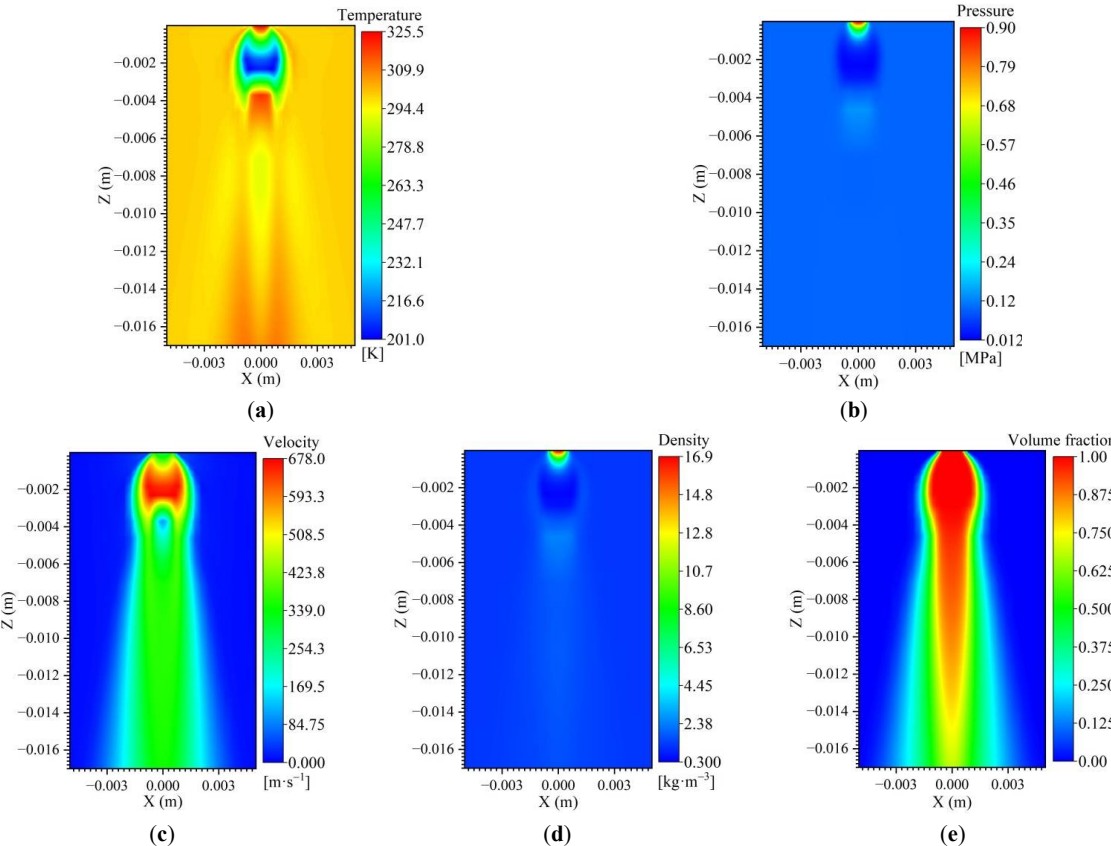

**Figure 8.** Cloud map of the field intensity in the local flow area below the leakage hole: (**a**) temperature cloud map; (**b**) pressure cloud map; (**c**) velocity cloud map; (**d**) density cloud map; (**e**) concentration cloud map.

From Figure 8a–c, it can be seen that after the leaked gas entered the external space, it firstly formed an annular cloud under the leakage hole, and then began to spread to the far-field flow area. The temperature and pressure of the gas at the center of the cluster were the smallest, and the velocity was the largest. In the horizontal direction, the gas pressure from the inner to the outer in the cloud cluster gradually increased to the ambient pressure, the temperature distribution was similar to the pressure distribution and the isotherm in the flow area below the cloud cluster presented an M-shaped distribution. The reason for this was mainly that the continuous expansion and compression processes occurred only in the gas near the jet axis in the flow area, whereas the gas near the jet boundary would not undergo further expansion and compression because its pressure was very close to the ambient pressure, which led to the result that the temperature of the compressed gas at the jet axis was higher than that of the gas between the jet axis and the jet boundary.

In the local flow area, when the temperature and pressure of the leakage gas changed, the density of the gas changed accordingly. From Figure 8d, it can be seen that the density of the mixed gas of leaked R290 and air was small at low pressure, and the density of the mixed gas was high at high pressure. In the horizontal direction, the density of the mixed gas from the inside to the outside gradually returned to the density of the gas at room temperature and normal pressure. The density of the mixed gas in the annular cloud cluster was the smallest, which could reach $0.34$ kg·m$^{-3}$, and the density of the mixed gas increased sharply through the Mach disk. The concentration field of the leaked gas in the local flow area is also shown in Figures 7d and 8e, and it can be seen that the R290 concentration was higher in the center of the annular cloud, the gas concentration in a small range along the jet direction did not decay and the gas concentration decreased rapidly through the Mach disk. In the horizontal direction, the gas concentration decreased from the jet axis to the jet boundary layer. Therefore, it can be known that the expansion and compression processes of the under-expanded R290 in the local flow area led to the deviation in the gas diffusion position away from the leak hole, which made the area of the high concentration region formed by the leaked R290 increase, and the formation of the Mach disk was not conducive to the diffusion of the leaked R290.

### 3.2. Concentration Distribution Characteristics of Leaked R290 in the Far-Field Flow Area

The variations in the R290 concentration with time at different locations in the far-field flow area is shown in Figure 9. The flow field distribution of the leaked R290 gas in the external limited space is shown in Figure 10. It can be seen from Figure 9 that with the increase in the leakage time, the gas concentration of each measuring point increased rapidly and then tended to be stable. In the direction of the jet, as the distance from the measuring point to the leakage hole increased, the concentration response time (the time from the beginning of the leakage to the concentration increase at the measurement point) was prolonged, and the concentration of R290 decreased. The concentration response times of points 7 and 11 were 0.06 and 0.2 s, respectively. After the concentration field in the local flow area was stabilized, the R290 concentrations of the points 3, 4, 7 and 11 were 86.5, 1.67, 0.85 and 0.56 vol%, respectively. Because the distance from point 3 to the leak hole was small, the concentration was higher, which was in the potentially flammable area, whereas at points 4, 7 and 11, the gas concentration had rapidly decayed to a lower level, which was in the nonflammable area. In the horizontal direction, the concentration response times of each measuring point was longer, and the R290 concentration was maintained at a low level. The concentration response times of points 8 and 9 were approximately 0.16 s, and the concentration values were relatively close, approximately 0.2 vol%. The concentration response times of points 10 and 12 were approximately 0.28 s, and the concentrations were maintained at 0.3 vol%. As the leaked R290 in the vertical direction was greatly affected by gas expansion process, the diffusion velocity of the gas was larger, the diffusion in the horizontal direction was mainly driven by the concentration gradient, and the diffusion velocity of the gas in the horizontal direction was much smaller than that in the vertical direction; thus, the concentration response time of each measuring point in the horizontal

direction was longer. In addition, it can be seen from Figure 10 that the concentration distribution of the leaked R290 in the horizontal direction was centered on the jet axis and symmetrically distributed in the circumferential direction.

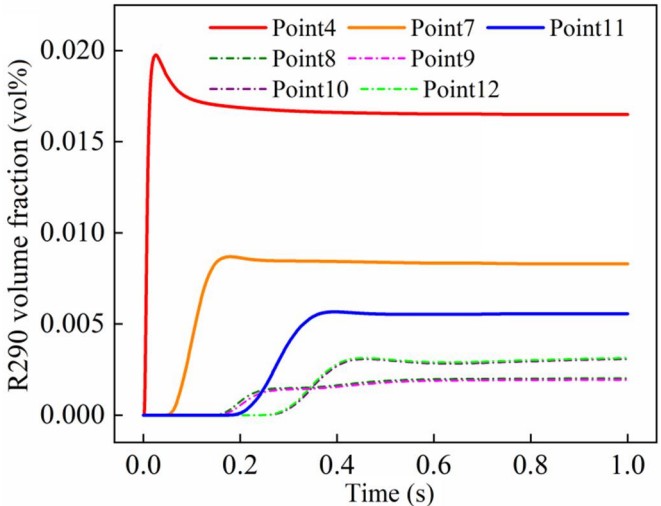

**Figure 9.** Changes in the R290 concentration at different locations in space over time.

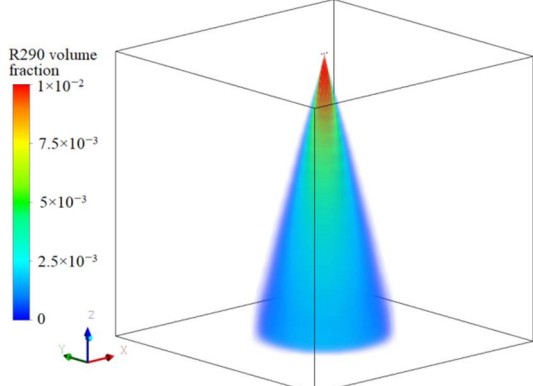

**Figure 10.** Flow field distribution of the leaked R290 gas in external space.

### 3.3. Effects of Different Initial Pressures on the Field Strength Characteristics of the Leaked R290

Figure 11 shows the pressure, temperature and velocity distributions of the leaked gas on the jet axis of the local flow area under different initial leakage pressures. It can be seen from Figure 11 that with the increase in the initial leakage pressure, the gas pressure at the outlet plane of the leakage section increased. When the initial pressures were 0.5, 1, 1.5 and 2 MPa, respectively, the pressures of the gas at the outlet of the leakage section reached 0.23, 0.46, 0.7 and 0.95 MPa. The leakage mass flow rates also increased, which were 0.84, 1.7, 2.6 and 3.6 g/s, respectively. The leaked R290 at a higher initial leakage pressure could expand to a lower pressure and temperature in the external space. When the initial pressures were 0.5 and 2 MPa, the minimum pressure and temperature of the gas in the annular cloud below the leak hole reached 0.046 MPa and 281.4 K and 0.013 MPa and 199.8 K, respectively; the temperature and pressure of the leaked gas at a high initial leakage pressure were higher after through the Mach disk. The reason for this was mainly that with the increase in the initial leakage pressure, the intensity of the shock wave generated by the leaked gas inside the near-field flow area increased, and a stronger driving force was generated during the gas expansion and compression process. At the same time, under high-pressure conditions, a higher velocity could be obtained during the expansion of the leaked gas. When the initial leakage pressures were 0.5 and 2 MPa, the maximum velocities of the leaked gas were 513.2 and 680.3 m/s, respectively. Moreover, after the gas expansion

and compression process disappeared, the leaked gas at a higher initial leakage pressure had a higher temperature and velocity on the jet axis.

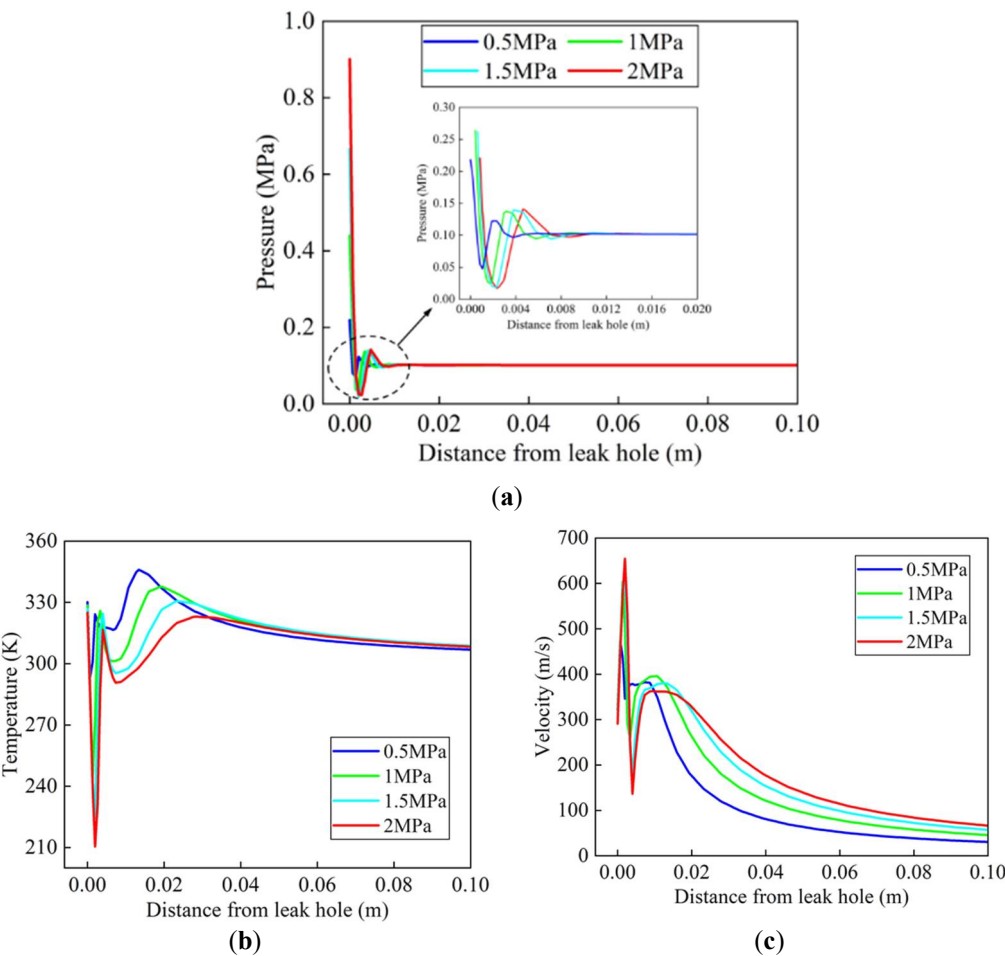

**Figure 11.** Field strength distribution of R290 on the jet axis under different initial leakage pressures: (**a**) pressure distribution; (**b**) temperature distribution; (**c**) velocity distribution.

There is a complex shock wave (Mach disk) structure inside the annular cloud formed by the leaked gas in the local flow area, which has a great impact on the concentration distribution of the leaked R290 gas. Therefore, it is very important to understand the position and circumferential diameter of the Mach disk under different initial pressures. As shown in Figure 12, it can be seen that as the initial leakage pressures increased from 0.5 to 2 MPa, the distances from the Mach disk to the leak hole were about 1.7, 2.3, 2.9 and 3.5mm, respectively, and the circumferential diameters of the Mach disk increased accordingly, but the number of the Mach disks did not change significantly. The reason for the above phenomenon was mainly that with the increase of the initial leakage pressure, the potential energy and kinetic energy of the leaked gas increased, but more gas accumulated under the leak hole during the same leakage time, and the jet boundary area increased. The energy loss of the gas during the expansion and compression process increased. After the leaked gas underwent two expansion and compression processes, the pressure was very close to the atmospheric pressure, so the number of Mach disks formed in the flow area did not change with the increase in the initial pressure. The increase in the distance from the Mach disk to the leak hole and the circumferential diameter of the Mach disk can be explained that the intensity of the expansion wave generated inside the annular cloud increased, which made the gas in expansion process reach a lower temperature and pressure, the expansion duration prolonged. At the same time, due to the increase of the flow area, it took longer time for the expansion wave to reach the jet boundary, and the low-pressure

gas inside the cloud could be jetted to a farther position, so a more intense shock wave (Mach disk) appeared farther away from the leak hole.

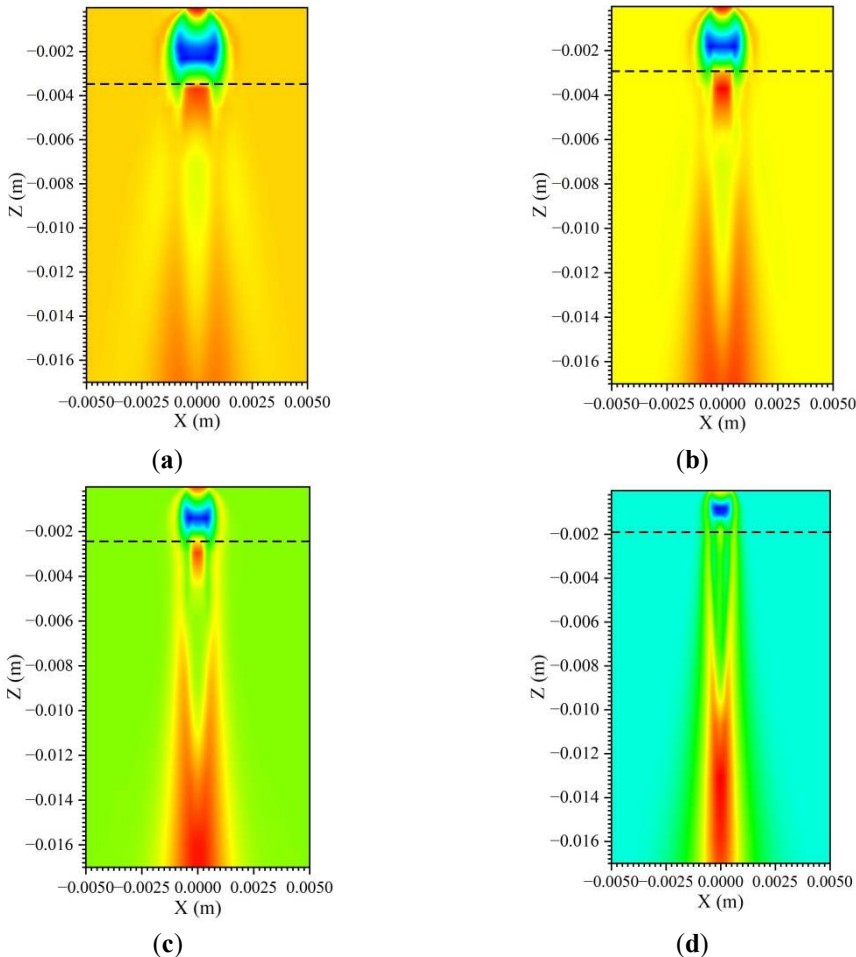

**Figure 12.** Variations in the Mach disk distance and circumferential diameter in the near-field flow area under different leakage pressures: (**a**) 2; (**b**) 1.5; (**c**) 1; (**d**) 0.5 MPa.

The curves of the R290 concentration changes over time at the measurement points in the external space under different pressures is shown in Figure 13, and the change in the flammable area formed by the leaked R290 in the external space under different leakage pressures is shown in Figure 14. It can be seen from Figure 13 that as the initial leakage pressure increased from 0.5 to 2 Mpa, the response times of the concentration change at the different locations in the space shortened, and the concentration values increased. After the local flow area concentration field stabilized, the concentrations of point 4 were maintained at 0.84, 1.12, 1.41 and 1.65 vol% under different pressures, the concentrations of point 7 were maintained at 0.42, 0.57, 0.71 and 0.83 vol% and the concentrations of point 11 were maintained at 0.28, 0.38, 0.47 and 0.56 vol%. The concentration response times of points 7 and 11 were shortened from 0.18 to 0.05 s and from 0.5 to 0.2 s, respectively. With the increase in the distance from the measuring point to the leak hole on the jet axis, the influence of the initial leakage pressure on the leakage R290 concentration decreased. It can be seen from Figure 14 that the flammable area formed by the leaked R290 was mainly concentrated in the local area below the leakage hole. As the initial leakage pressure increased from 0.5 to 2 Mpa, the ranges of the flammable area in the horizontal direction changed little, whereas the ranges of the flammable areas in the vertical direction increased significantly: 0.2, 0.27, 0.34 and 0.4 m. The reason for this was mainly that with the increase in the initial pressure, more R290 leaked into the external space and diffused to different positions at a higher velocity, which made the response time of the concentration change

shorten. The gas diffusion velocity in the vertical direction was much greater than that in the horizontal direction, resulting in a higher concentration gradient of the leaked gas in the vertical direction.

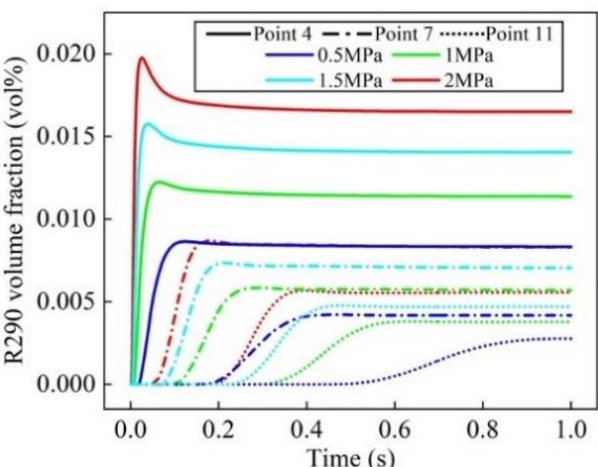

**Figure 13.** Variations in the R290 concentration in external space over time under different initial leakage pressures.

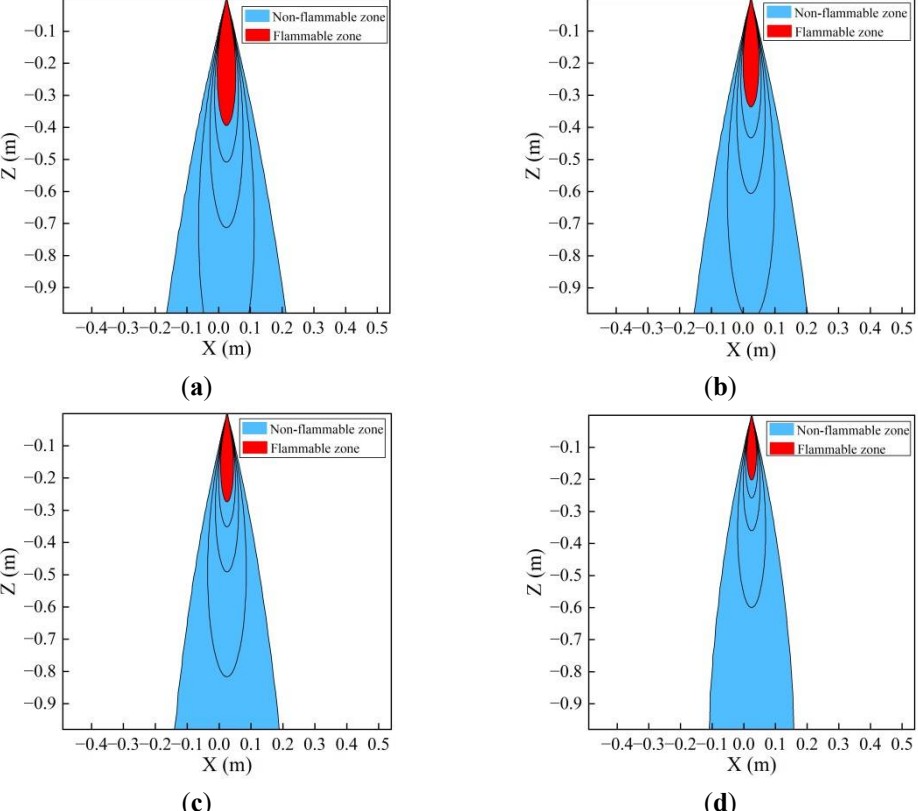

**Figure 14.** Variations in the flammable area of the external space under different initial leakage pressures: (**a**) 2; (**b**) 1.5; (**c**) 1; (**d**) 0.5 MPa.

## 4. Conclusions

In this paper, the refrigerant leakage process based on the refrigeration system was analyzed in detail, and an improved gas leakage mode was proposed to describe the process of the refrigerant leakage and diffusion. Taking R290 as the refrigerant, the change law of the field strength of the leaked R290 and the mutual coupling characteristics between

multiple fields were discussed when R290 leaked from the refrigeration system into the external space, and the influence of different initial leakage pressures on the field strength of the leaked R290 was analyzed. The conclusions are as follows:

(1) An improved gas leakage model suitable for describing refrigerant leakage in the refrigeration system was proposed, and the calculation results obtained by the improved model were compared with the experimental data from the published literature. The results found that the theoretical values of the position and circumferential diameter of the Mach disk and the concentration of the leaked gas were in good agreement with the experimental values, and the reliability of the improved gas leakage model was verified;

(2) Based on the improved leakage model, the leakage and diffusion characteristics of R290 were analyzed. When the initial pressure was 2 MPa, the temperature was 360 K, the R290 gas continued to expand after it leaked into the external space, and its temperature and pressure decreased rapidly, reaching the minimum of 199.8 K and 0.013 MPa, respectively; its velocity increased rapidly, and the maximum reached 680.3 m/s. A Mach disk was formed in the flow area below the leakage hole, and the R290′s thermo-physical parameters changed abruptly through the Mach disk. After two consecutive expansion and compression processes, the temperature, pressure and the velocity of the R290 gas gradually decreased along the jet direction and, finally, the R290 gas returned to room temperature and normal pressure. In addition, the expansion and compression process of the leaked R290 gas in the local flow area caused the gas diffusion position to deviate away from the leakage hole, and the high concentration area formed by R290 increased. The formation of the Mach disk was not conducive to the diffusion of the leakage gas;

(3) The initial leakage pressure had a great influence on the position and circumferential diameter of the Mach disk formed by the leaked gas. As the initial leakage pressure increased from 0.5 to 2 MPa, the distances from the Mach disk in the local flow area to the leakage hole were approximately 1.7, 2.3, 2.9 and 3.5 mm, and the circumferential diameters of the Mach disk increased accordingly. With the increase in the initial leakage pressure, the concentration response times of different measuring points were shortened (For example, the response times of the points 7 and 11 were shortened from 0.18 to 0.05 and 0.5 to 0.2 s, respectively). The flammable area formed by the leaked R290 in the external space slightly increased in the horizontal direction, the flammable area increased significantly in the vertical direction, and its ranges were within 0.2, 0.27, 0.34 and 0.4 m, and the flammable area was mainly concentrated in the local area below the leakage hole.

The above research suggests that the improve leakage model will provide a valuable reference to predict the flammable area.

**Author Contributions:** Conceptualization, formal analysis, investigation, and methodology, Y.L. (Yalun Li); Resources, P.Z., Y.Z. and X.H.; Supervision, X.W., Y.L. (Ying Liu), X.H. and G.C.; Writing—original draft, Y.L. (Yalun Li); Writing—review and editing, X.H. All authors have read and agreed to the published version of the manuscript.

**Funding:** This research was supported by the National Natural Science Foundation of China (Grant Nos. 51936007 and 52076185).

**Data Availability Statement:** All the data analyzed in the article have been given.

**Acknowledgments:** The authors of this paper thank the National Natural Science Foundation of China (Grant Nos. 51936007 and 52076185) for funding.

**Conflicts of Interest:** The authors declare no conflict of interest.

## Nomenclature

| | |
|---|---|
| $Q$ | leakage mass flow rate (kg s$^{-1}$) |
| $A_{or}$ | leak hole area (m$^2$) |
| $p$ | pressure (MPa) |
| $T$ | temperature (K) |
| $k$ | isentropic index |
| $M$ | kilomolar mass (kg kmol$^{-1}$) |
| $R$ | gas constant (kJ kmol$^{-1}$ K$^{-1}$) |
| $G$ | mass flow rate per cross-sectional area (kg m$^{-2}$ s$^{-1}$) |
| $Z$ | compressible factor |
| $f$ | friction coefficient of the pipe wall |
| $D$ | pipe diameter (m) |
| $L_e$ | effective pipe length (m) |
| $c$ | sound velocity (m s$^{-1}$) |
| $Ma$ | Mach number |
| $c_P$ | heat capacity (kJ kg$^{-1}$ K$^{-1}$) |
| $k_t$ | turbulent thermal conductivity (kW m$^{-1}$ K$^{-1}$) |
| $D_t$ | turbulent mass diffusion coefficient (m$^2$ s$^{-1}$) |
| $V$ | specific molar volume (m$^3$ kmol$^{-1}$) |
| $g$ | acceleration of gravity (m s$^{-2}$) |

Greek symbols

| | |
|---|---|
| $\xi$ | orifice loss coefficient |
| $\rho$ | density (kg m$^{-3}$) |
| $\mu$ | viscosity (kg m$^{-1}$ s$^{-1}$) |
| $\omega$ | mass fraction |
| $\sigma$ | L-J characteristic length (angstrom) |

Subscripts

| | |
|---|---|
| $a$ | air |
| $cr$ | critical |
| $e$ | effective |
| $r$ | refrigerant |
| $t$ | turbulence |
| max | maximum |
| $i, j$ | unit vector direction |
| $ci$ | number of components |

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
