# Peer review of "An Improved Gas Leakage Model and Research on the Leakage Field Strength Characteristics of R290 in Limited Space"

_applsci, doi:10.3390/app12115657_

Round 1
Reviewer 1 Report
A nomenclature is to be added.
Several assumptions are considered in this study, without presenting the justifications.
The authors studied numerically the considered configuration, what the interest of the long theoretical part present in the preceding parts.
The authors used ANSYS software in this study without presenting the governing equations.
The implemented boundary conditions are to be expressed mathematically.
Have you considered turbulent or laminar regime?
The title of Fig 6 is very confusing, is it ‘’numerical’’ or ‘’theoretical’’?
The authors studied a 3D configuration without presenting any 3D profile.
A grid sensitivity test is to be performed.
A qualitative validation/verification is to be performed.
The velocity field at the pipeline is to be presented.
The English level of the paper is generally good, but there are some misprints and grammatical mistakes.
Author Response
Thank you so much for the valuable comments, which are of great help in improving the quality of this manuscript. We have carefully revised the manuscript. The revised parts are marked in blue in the paper, and the responding answers to these suggestions are shown point by point, as follows:
Comment 1: A nomenclature is to be added.
√Thank you for your comments.
We have added nomenclature in the paper, and the details can be seen in the revised manuscript.
Comment 2: Several assumptions are considered in this study, without presenting the justifications.
√Thank you for your comments.
Some considerations in establishing these assumptions have been added in the paper, and the details can be seen in Section 2.3.3 and marked in blue of the revised manuscript.
Comment 3: The authors studied numerically the considered configuration, what the interest of the long theoretical part present in the preceding parts.
√Thank you for your comments.
The refrigerant leakage characteristics in refrigeration systems usually were analyzed by means of the hole model in many literatures.
The hole model reveals the leakage mechanism of the gas in the pipeline to a certain extent, but there are still limitations: 1) The gas flow state in the interval from the leakage port in the pipeline to the wall of the pipeline during the gas leakage process is ignored in the modeling. 2) The flow state of the gas in the leakage section is directly determined according to the given critical pressure ratio formula, and the pressure of the gas at the outlet of the leakage section is assumed. For example, when p1 (the pressure inside the pipeline)<pcr (the critical pressure, that is the corresponding pressure inside the pipeline when the gas in the leakage section changes from subsonic flow to sonic flow), it is considered that the pressure of the gas at the leak hole is equal to the atmospheric pressure, p2=pa; when p1≥pcr, it is considered that the pressure of the gas at the leak hole is equal to p1·CPR (Critical pressure ratio), p2= p1·CPR. The above limitations will lead to certain deviations in the calculated leakage mass flow rate and state parameters of leaked gas.
In this work, an improved leakage model is proposed, the comparison of the current hole model and the improved gas leakage model is as shown in Table 1 of the revised manuscript. By the improved leakage model, the more accurate leakage mass flow rate and leakage gas state parameters can be obtained under all hole leakage conditions, and the Mach disk formed by the leaked gas in the local flow area of the external space can be found, and the real gas model-PR equation is used in the model to calculate the parameters of the leaked refrigerant gas, such that the range of the flammable area formed by the leaked refrigerant in the external space can be more accurately predicted. At the same time, the improved model can be used to provide guidance for the theoretical calculation of refrigerant leakage.
Comparison of the current hole model and the improved leakage model was shown Table 1.
Table1 Comparison of the current hole model and the improved leakage model
Model Leakage process |
The current hole model |
The improved gas leakage model |
|
The schematic figure |
|||
Similarity |
The transport process 0-1 |
The same model and solution method were used. |
|
Difference |
The leakage process 1-2 |
The gas pressure at the outlet of the leakage section was directly assumed based on the given critical pressure ratio formula. |
The relationship between the gas flow parameters and the flow section was established, and the gas state parameters at point 2 can be solved. |
The expansion process 2-3 |
The change of the gas state parameters in the interval from the leakage port in the pipeline to the outer wall of the pipeline was ignored. |
The expansion process of the gas in the leakage section was considered, and the state parameters of the gas at the outlet of the leakage section can be obtained. |
Comment 4: The authors used ANSYS software in this study without presenting the governing equations.
√Thank you for your comments.
The theoretical analysis and corresponding control equations of the gas leakage process can be found in Section 2.2, and we have supplemented the control equations of the diffusion process of the leaked gas in the external space, which can be seen in Section 2.3.1 of the revised manuscript.
Comment 5: The implemented boundary conditions are to be expressed mathematically.
√Thank you for your comments.
The solution methods and initial values of some parameters in the numerical model are made up, and the details can be seen in Table 3 of the revised manuscript.
Table 3. The solution method or initial value of parameters in the numerical model
Numerical model |
Method or initial value |
Turbulence model |
SST k-omega |
Leakage gas |
R290 |
Density |
|
Viscosity |
|
Thermal conductivity |
|
Diffusion coefficient |
|
Initial leakage pressure P0 (MPa) |
0.5, 1, 1.5 and 2 |
Initial leakage temperature T0 (K) |
360 |
Pressure of the external space Pa (kpa) |
101.325 |
Temperature of the external space Ta (K) |
300 |
Solver |
Pressure-based, transient |
Solution method |
Coupled |
Note: a and b are empirical constants, which can be obtained from literature [34],ΩD is a dimensionless parameter, which can be obtained from [35]。
Comment 6: Have you considered turbulent or laminar regime?
√Thank you for your comments.
The flow state of the leaked gas has been considered.
During the actual process of the gas with a certain pressure leaking to the external space through the leak hole, the leaked gas in the pipeline, the leakage section and the local flow area of the external space generally has a high velocity, and the gas may even flow at the velocity of sound, and the Reynolds number is relatively high [1-3]. Moreover, it is found from the simulation results in the paper that when the initial leakage pressure is 2MPa, the velocity of the R290 gas at the inlet of the leakage section is 150.9 m/s, and the gas is still in a subsonic flow state, the Reynolds number is about 4.5×105. The gas velocity at the outlet of the leakage section reaches 272.15 m/s, the gas is in the sonic flow state, and the Reynolds number is about 5.2×105. After the leaked gas enters the external space, the velocity further increases due to the continuous expansion process. Therefore, the leaked gas is considered in a turbulent state in the process of refrigerant leakage in this paper.
[1] Lin T, Li Y, Hu Q, Zhang D, Xiao Y, Gu S, Wang C. Experimental study of near-field structure and thermo-hydraulics of supercritical CO2 releases[J]. Energy, 2018, 157, 806-814.
[2] Li X, Wu K, Yao W, X Fan. A comparative study of highly underexpanded nitrogen and hydrogen jets using large eddy simulation[J]. International Journal of Hydrogen Energy, 2016, 41, 5151-5161.
[3] Li X, Zhou R, Yao W, X Fan. Flow characteristic of highly underexpanded jets from various nozzle geometries[J]. Applied Thermal Engineering, 2017, 125, 240-253.
Comment 7: The title of Fig 6 is very confusing, is it ‘’numerical’’ or ‘’theoretical’’?
√Thank you for your comments.
We have modified the title of Figure 6, and the expression is as follows: Figure 6. Comparison of experimental and numerical results on the concentration of leaked hydrogen.
Comment 8: The authors studied a 3D configuration without presenting any 3D profile.
√Thank you for your comments.
A three-dimensional cloud map has been added in the paper, the details as shown in Figure 10 of the revised manuscript. Most of the maps in the paper are represented in two dimensions, mainly because we consider that after the gas in the pipeline leaks the external space through a circular leak hole, the flow field distribution of the leaked gas is symmetrically distributed in the circumferential direction with the jet axis as the center, so one of the cross sections is selected as an example to analyze, and the 2D cloud map can better capture the Mach disk structure inside the flow area.
Figure 10. Flow field distribution of leaked R290 gas in external space
Comment 9: A grid sensitivity test is to be performed.
√Thank you for your comments.
Before proceeding with the work in this paper, we have considered the influence of the mesh type and mesh quantity, and we performed numerical simulations for unstructured meshes (80W) and structured meshes (80W and 150W), respectively. The comparison between the numerical simulation results of structured grid (80W) and unstructured grid (80W) and the experimental data of Section 2.4 in this paper is shown in below Figure 1. It can be seen from Figure 1 that the results obtained with the structured grid (80W) under different pressure conditions deviate less from the experimental values. The comparison of the state parameters at measuring point 2 obtained by the structured grid (80W) and the structured grid (150W) is shown in below Table 1. It can be seen from the table that the deviation of the results is small under the grid quantities of 80W and 150W. Therefore, through comprehensively comparing the results deviation, convergence accuracy and calculation velocity, the structured mesh (80W) is selected for the work of this paper. Considering that the focus of this paper is to analyze the field strength of the leaked gas, this part of the content is not presented in the paper.
Figure 1. Comparison of theoretical and experimental values of leaked hydrogen concentration for different grid types
Table 1. Comparison of calculation results with different grid numbers
Mesh (quantity) |
Pressure (MPa) |
Temperature (K) |
Velocity (m/s) |
Mass flow rate (g/s) |
Structured mesh (150W) |
0.399 |
234.32 |
1371.85 |
0.41 |
Unstructured mesh (150W) |
0.401 |
233.46 |
1378.04 |
0.42 |
Comment 10: A qualitative validation/verification is to be performed.
√Thank you for your comments.
In this paper, an improved gas leakage model is established, and it can be used for the ideal gas and the real gas by means of the thermo-physical properties of the working fluids.
Hydrogen is selected to verify the reliability of the established model, the reason is mainly that the hydrogen leakage is widely investigated in the published literatures, and there are many experimental data, and the Mach disk formed by the leaked gas has been found in the flow area, which can be used to verify the reliability of the established model to a certain extent. In the following numerical simulation of R290 leakage, the difference between the thermo-physical properties of R290 and hydrogen is also considered, so the real gas model-PR equation is used in the model to calculate the parameters of the leaked R290 gas. Under different temperature and pressure conditions, the compared results of the density of R290 based on the PR equation and Refprop are shown in the table 2. It can be seen from the table that the density of R290 calculated by the PR equation has a small deviation from the actual value. Therefore, the simulation results in this paper are reliable to a certain extent.
Table 2. Comparison of calculation results of R290 density under different conditions
Pressure /MPa |
Temperature /K |
Density kg/m3 |
Deviation/% |
|
Refprop |
PR equation |
|||
2 |
335 |
44.215 |
44.789 |
1.3 |
340 |
42.371 |
43.049 |
1.6 |
|
345 |
40.786 |
41.514 |
1.78 |
|
350 |
39.395 |
40.140 |
1.89 |
|
355 |
38.153 |
38.898 |
1.95 |
|
360 |
37.032 |
37.766 |
1.98 |
|
1.5 |
325 |
31.408 |
31.708 |
0.96 |
330 |
30.38 |
30.729 |
1.15 |
|
335 |
29.456 |
29.834 |
1.28 |
|
340 |
28.617 |
29.010 |
1.37 |
|
345 |
27.849 |
28.248 |
1.43 |
|
350 |
27.14 |
27.539 |
1.47 |
|
1 |
305 |
21.024 |
21.086 |
0.29 |
310 |
20.416 |
20.514 |
0.48 |
|
315 |
19.858 |
19.983 |
0.63 |
|
320 |
19.344 |
19.487 |
0.74 |
|
325 |
18.867 |
19.021 |
0.82 |
|
330 |
18.421 |
18.584 |
0.88 |
Comment 11: The velocity field at the pipeline is to be presented.
√Thank you for your comments.
When the initial leakage pressure is 2MPa, the velocity distribution in the pipeline and the leakage section is shown in below Figure 2. Under different initial leakage pressures, the velocity of the gas in the pipeline and the leakage section changes greatly. To monitor the change of gas velocity in the leakage section, two measuring points have been set at the inlet and outlet planes of the leakage section in the geometric model in the paper. But, in this work, we focus on the field strength change of the flammable refrigerant and the distribution characteristics of the flammable area after the refrigerant leaks into the external space.
Figure 2. Cloud map of velocity distribution in the pipeline
Comment 12: The English level of the paper is generally good, but there are some misprints and grammatical mistakes.
√Thank you for your comments.
The sentences and grammar in the manuscript have been revised carefully, and details can be seen the contents marked in blue words in the revised manuscript.

Reviewer 2 Report
The subject of the paper is really interesting, and I suggest to publish the paper after considering the following comments:
1- More detail is presented about the numerical model used to solve the governing equations.
2- Remove the legends in Fig. 7.
Author Response
Thank you so much for the valuable comments, which are of great help in improving the quality of this manuscript. We have carefully revised the manuscript. The revised parts are marked in blue in the paper, and the responding answers to these suggestions are shown point by point, as follows:
Comment 1: More detail is presented about the numerical model used to solve the governing equations.
√Thank you for your comments.
We have added the introduction of the solving methods of the numerical model, as the details can be seen in Section 2.3.3 of the revised manuscript
Comment 2: Remove the legends in Fig. 7.
√Thank you for your comments.
The legend of Figure 7 has been removed, and the details can be seen in Fig. 7 of the revised manuscript.
Reviewer 3 Report
In the manuscript the leakage and diffusion characteristics of the refrigerants is discussed proposed in this paper with some experimental verification. The idea is interesting and promising, but the description has to be improved. First there is no real description of the test stand, measurement devices, uncertainty, no drawing or picture of the test stand. The model is for ideal gas, the validation is for hydrogen this is probably OK, but for R290 using ideal gas assumption may generate significant errors.
Author Response
Thank you so much for the valuable comments, which are of great help in improving the quality of this manuscript. We have carefully revised the manuscript. The revised parts are marked in blue in the paper, and the responding answers to these suggestions are shown point by point, as follows:
Firstly, we select the experimental data in the published literature to verify the reliability of the established model in this paper, and the detailed description of the experimental device and measurement equipment is not presented in the paper, the details can be seen in section 2.4 literature [36]. Secondly, regarding the model established in this paper, the ideal gas equation is adopted to simplify the derivation during the modeling process, and we have considered the difference between the properties of the refrigerant used in the refrigeration system and the ideal gas, so the compression factor Z is introduced to reduce the deviation caused by the calculation of the actual gas leakage. To verify the reliability of the established model, hydrogen is selected, and the ideal gas state equation is used to calculate the thermo-physical parameters of leaked hydrogen. However, in the following numerical simulation of R290 leakage, we use the real gas model-PR equation to solve the thermo-physical parameters of leaked R290. Under different temperature and pressure conditions, the compared results of the R290 density obtained by the ideal gas state equation, PR equation and Refprop are shown in the below table. It can be seen from the table that the density of R290 calculated by the PR equation has a small deviation from the actual value, so the simulation results in this paper are also reliable to a certain extent. The description of the calculation method in the paper is incorrect, and we have revised it, the details can be seen in Section in Section 2.2 and 2.3.3 of the revised manuscript.
Table 3. Comparison of calculation results of R290 density under different conditions
Pressure /MPa |
Temperature /K |
Density kg/m3 |
||
Refprop |
Ideal gas equation |
PR equation |
||
2 |
335 |
44.215 |
31.667 |
44.789 |
340 |
42.371 |
31.201 |
43.049 |
|
345 |
40.786 |
30.749 |
41.514 |
|
350 |
39.395 |
30.310 |
40.140 |
|
355 |
38.153 |
29.883 |
38.898 |
|
360 |
37.032 |
29.468 |
37.766 |
|
1.5 |
325 |
31.408 |
24.481 |
31.708 |
330 |
30.38 |
24.110 |
30.729 |
|
335 |
29.456 |
23.750 |
29.834 |
|
340 |
28.617 |
23.401 |
29.010 |
|
345 |
27.849 |
23.062 |
28.248 |
|
350 |
27.14 |
22.732 |
27.539 |
|
1 |
305 |
21.024 |
17.391 |
21.086 |
310 |
20.416 |
17.110 |
20.514 |
|
315 |
19.858 |
16.839 |
19.983 |
|
320 |
19.344 |
16.575 |
19.487 |
|
325 |
18.867 |
16.320 |
19.021 |
|
330 |
18.421 |
16.073 |
18.584 |
In the future, the more experimental tests will be developed, and the more analyses will be presented. Thank you so much for your valuable comments!

Round 2
Reviewer 1 Report
After revision the paper can be accepted for publication
Reviewer 3 Report
The manuscript is improved on the satisfactory level. I had a wrong impression that Authors did the experimental work by themselves, but in case of literature experimental data the description is ok.